# Neural Disambiguation of Causal Lexical Markers based on Context

## Abstract

Causation is a psychological tool of humans to understand the world and it is projected in natural language. Causation relates two events, so in order to understand the causal relation of those events and the causal reasoning of humans, the study of causality classification is required. Herein, we propose a neural network architecture for the task of causality classification. We claim that the encoding of the meaning of a sentence is required for the disambiguation of its causal meaning. Our results show that our claim holds, and we outperform the state-of-the-art.

## 1 Introduction

Causation is a psychological tool of humans to understand the world independently of language, and it is one of the principles involved in the construction of the human mental model of reality (Neeleman and van de Koot, 2012). Causal reasoning is the process of relating two events, namely cause and its effect. Following the words of Reinhart (2002), causal relations are imposed by humans on the input from the world, and the (computational) linguist's task is to understand what is about language that enables speakers to use it to describe their causal perceptions.

There are different theories concerning how natural language approximates causation. Those theories based on generative semantics argue that the causal relation is encoded in the semantics of some verbs (Lakoff, 1970; McCawley, 1976) or in the syntactic structure of a sentence (Dowty, 1979; Ramchand, 2008). For instance, Pylkkänen (2008) argues that the causing event may be associated with the subject of the causal verb in a causal predicate. This principle is true in the sentence "*Pe-*

*ter eventually killed John by hitting him with a hammer*", in which the event of Peter hitting John with a hammer caused the death of John. But, the following sentence can be used as a counterexample: "*A hammer eventually killed John*". In the later sentence the subject of "*killed*" is the "*hammer*", which does not constitute a causing event for the event "*kill John*". Therefore, the syntax-grounded construction of causality defined by those theories is far away from the human mental model of causation. Indeed, Neeleman and van de Koot (2012) showed that the later approaches cannot be proven by standard syntactic tests, because neither the causing event nor causal relation correspond to a syntactic constituent. Neeleman and van de Koot (2012) concluded that the linguistic approximation of causation by culmination of events is not exclusive of causal verbs, and it is also found in non-causative verb classes.

Causation or causality has also been studied in computational linguistics. There are some semantic and discourse resources that take into account causality in the range of linguistic phenomena that they annotate. PropBank (Palmer et al., 2005) is a semantic resource that annotates the argument structure of verbs. The type of causation annotated in PropBank is related to the syntax and semantics of the verbs. For the sentences *John broke the window* and "*The hammer broke the window*", "*John*" and "*the hammer*" are the cause arguments of the verb "*broke*". Although, "*John*" and "*the hammer*" are required arguments for the event expressed by "*break*", they do not represent the causing event of the window breaking event.

Causality is one of the discourse relations in Penn Discourse Tree Bank (PDTB) (Prasad et al., 2008). Causality is annotated in PDTB as either an explicit or an implicit relation between events. When causality is explicit, it is signalled by a lexical marker such as *because* or *since* between others.

However, there are some scenarios where the causal discourse relation is not signalled by a known lexical marker, but by other expressions that are called Alternative Lexicalizations (AltLex). The existence of AltLex means that the expression of causality is not defined by a limited number of lexical constructions, so the coverage of the total number of possible causal expressions is restricted to the grade of coverage of the base corpus of PDTB.

The previous paragraphs and the works described in Section 2 show that causality is driven by a limited set of lexical elements and syntactic constructions. However, causality does not have a fixed list of lexical or syntactic constructions. Hidey and McKeown (2016) stressed the issue of the limited coverage of causal expressions and released a corpus with a larger amount of causal expressions than PDTB. They also proposed a classification method, which is based on the use of features from the corpus and features from lexical resources. The last set of features reduces the ability of the system to classify causality, because they restrict the system to the causal definition of the lexical resources.

Causal meaning disambiguation is the task of identifying whether there is a causal relation between two events. We hypothesize that neural networks are able to encode the meaning of those events, and discover whether the underlying relation between them is causal. Herein, we contribute a neural network architecture for the task of causality disambiguation and we assess it in the corpus of Hidey and McKeown (2016). Empiric results on this corpus show that our claim indeed holds.

## 2 Related Work

Khoo et al. (1998) presented one of the first works related to the classification of causal relations. The authors defined a set of linguistic rules for the identification of cause-effect relations, and developed a pattern matching system based on those rules. The errors of the system show the common errors of a full linguistically grounded classification system, which are related to the ambiguity of some causal links such as *by* or *as* or the lexical and syntactic ambiguity of some causative verbs. Those errors are related to the lack of evidence of the syntax basis of causation (Neeleman and van de Koot, 2012).

Girju (2003) proposed a decision tree learning method to classify causal relations between events. The type of relations studied by the author are those conveyed by a verb and two noun phrases. The classification method uses as lexical features whether the verb of the sentence belongs to a list of ambiguous causative verbs, and whether the nouns belongs to one of the nine semantic hierarchies in WordNet. Again, the main source of errors is the ambiguity of the projection of causality in natural language.

Bethard and Martin (2008) defined a method for the classification of causal relations between verbal events in conjunction constructions, namely conjoined events. Since the method was defined for a specific kind of causal relation between events, the authors had to first create a corpus following that requirement. The authors developed a supervised classification system using syntactic and semantic features from WordNet, and they used SVM as a classification algorithm. The drawback of the method is its reduced flexibility, because it is only defined for a specific causal construction and its confidence in syntactic features. Moreover, the authors conclude that semantic features are more adequate for causality classification than syntactic features, which goes in the direction of our claim.

The first attempt of encoding the contextual information for causality classification between events is found in Do et al. (2011). The authors defined a measure of the causality relation between two events according to the co-occurrence frequency between the predicates and their arguments. The measure is inspired by the causality score defined by Riaz and Girju (2010).

Following the work of Do et al. (2011), Riaz and Girju (2013, 2014) built a knowledge base for causal events, first for events related by a pair of verbs, and then for events conveyed by pairs of verbs and noun phrases. In these works, the authors attempted to go in the direction of using distributional semantics in order to encode the causal meaning, but they still rely on the use of lexical and syntactic features. Thus, the main drawbacks of those papers are the complex featuring engineering process to detect the causal relation, and the restricted coverage of the resulting knowledge bases.

Following the previous approaches, Mirza and Tonelli (2016) described a causal classification system built upon a rule based system and a machine learning algorithm based on lexical, syntactic and semantic features. The novelty lies in the use of features of temporal relations. The work of Mirza and Tonelli (2016) has similar issues to the work of (Bethard and Martin, 2008), because they use similar features to represent causality.

| Sentence | Type |
|---|---|
| (Cathay Pacific delayed both legs of its quadruple daily Hong Kong to London route)$_{e_1}$ ([due to]$_l$ this disruption in air traffic services.)$_{e_2}$ | Explicit |
| (The factory was not well equipped to handle the gas)$_{e_1}$ (created by the sudden addition of water to the MIC tank.)$_{e_2}$ | Implicit |

Table 1: Explicit and implicit causal relations

The previous works support the idea that causality is defined by a specific syntactic construction or the semantics of verbs known as causative verbs. On the other hand, some of the works restricted their study to a narrow kind of syntactic constructions, such as the conjoined events of Bethard and Martin (2008). In contrast, we argue that the representation of causality should not be limited by syntactic patterns or the coverage of semantic resources, due to the lack of a particular linguistic construction for causality. Therefore, we propose the neural encoding of the context of the relation in order to disambiguate its causal meaning.

## 3 Causality classification

The next sections expose the definition of the task (Section 3.1), the data used in the experiments (Section 3.2) and the proposed system (Section 3.3).

### 3.1 Definition

Causality is defined as the semantic relation between two events $(e_1, e_2)$, namely causing event (cause, $e_1$) and caused event (effect, $e_2$). An event may be a verb, whose arguments may be explicitly present or not (*subject, verb, object*), or a noun phrase. The relation can be explicit or implicit, in case it is explicitly expressed, it is signalled by a lexical marker (*l*), which may be a verb, a preposition, an adverb or an expression such as *as a result of*, see Table 1.

Some researchers impose two additional restrictions: the temporal restriction, which means that the causing event should take place before the caused event, and the counterfactual one, which says if the causing event did not occur, the caused event would not have occurred either. Thus, causation can be formally defined as $e_1 \rightarrow e_2$. An example from the test set used in the experiments that follows the three restrictions is the following:

> *A government affidavit in 2006 stated that (the leak)$_{e_1}$ (caused 558,125 injuries, including 38,478 temporary partial injuries and approximately 3,900 severely and permanently disabling injuries)$_{e_2}$*

It is evident that $e_1$ is before $e_2$ and if the *leak* did not occur, the set of events in $e_2$ would not occur. Nevertheless, we support the same opinion as Neeleman and van de Koot (2012) and we do not add the counterfactual constraint in our definition, because it is not true in all the cases. For example:

> *(The argument between John's parents)$_{e_1}$ (broke the window)$_{e_2}$*

If we follow the counterfactual constraint, the event of the argument between the John's parents should be the causing event. However, the human mental model of causality does not usually relate an argument with the breaking of a window. Causation is concerned with the human understanding of the world and not with the world itself.

Concerning the temporal restriction, it should be interpreted as a physical ordering constraint and not as a positional one. That means, the causing event does not always appear before the caused event in the sentence. The caused event ($e_2$) is before the causing event ($e_1$) in the following example:

> *(This water was diverted)$_{e_2}$ ([due to]$_l$ a combination of improper maintenance, leaking and clogging, and eventually ended up in the MIC storage tank.)$_{e_1}$*

Some related work attempted to identify the causing, the caused event and its direction (Mirza and Tonelli, 2016), but for that purpose, they restricted the study to a specific syntactic construction. We claim that the task of causality classification has not to be restricted to a syntactic construction and it has to be set up as a two steps task: causal meaning classification and causal arguments identification. The task of causal meaning classification is a binary classification task that is defined as the disambiguation of the causal meaning of the relation of two events. The input of the task is two events and the output is the meaning of the relation (`Causal` or `Non Causal`). The task of causal argument identification is focused on the identification of the causing ($e_1$) and caused ($e_2$) events. The aim of this paper is to contribute to the first subtask, so given two events ($e_1$, $e_2$) and the expression susceptible of signalling causality (*l*), the system returns the meaning of the relation (`Causal` or `Non Causal`). The following sentence is an example of the input of our system.

> *(He fell in love with her and changed his life)$_{e_1}$ ([so]$_l$ he could help her)$_{e_2}$*

| Sentence | Meaning |
|---|---|
| An undercroft is traditionally a cellar or storage room, often brick-lined and vaulted, and used for storage in buildings *since* medieval times | Temporal |
| Additionally if one is to use a large scan range then sensitivity of the instrument is decreased due to performing fewer scans per second *since* each scan will have to detect a wide range of mass fragments | Causal |
| In stark contrast to his predecessor, five days *after* his election he spoke of his determination to do what he could to bring peace | Temporal |
| Bischoff in a round table discussion claimed he fired Austin *after* he refused to do a taping in Atlanta | Causal |

Table 2: Examples of different meanings of the prepositions *since* and *after* taken from the AltLex corpus

## 3.2 Data

Causality may be signalled by a verb (*cause*), a preposition (*because*), an adverb (*subsequently*) or an expression such as *as a result of*. Some of those causal expressions are almost unambiguous, but the causal meaning of others totally depends on the context. For instance, according to PDTB, the preposition *because* is an unambiguous lexical marker of causation, but there are other expressions whose causal meaning is not unarguable, such as *since* or *after* that may have a temporal or a causal interpretation, see Table 2. Therefore, we need a corpus in which the events, the lexical marker and the meaning of the relation are annotated.

To the best of our knowledge, there are three available corpora. The first one was exposed in Bethard and Martin (2008) and it is built on top of the Penn TreeBank corpus. The corpus only provides the position of the related events in each sentence and whether the relation is causal. Unfortunately, the corpus is not freely available, and is also not large enough to train neural methods.

The second corpus is Causal-TimeBank (Mirza et al., 2014), which is built on top of the TimeML corpus. Annotation was restricted to explicit causal relations. They followed the same annotation schema of TimeML, so they defined the label C-SIGNAL for the causal lexical markers, and C-LINK to mark the causal relation between two events. The authors manually annotated the causal relations between the identified events in TimeML. The fact that the causal lexical markers are annotated agrees with our data requirements, but as the previous corpus, Causal-TimeBank is too small to train neural methods.

Recently, the AltLex corpus has been freely released (Hidey and McKeown, 2016). The corpus was built on the idea that causation can be expressed by different kinds of linguistic constructions. This is validated by the fact that in PDTB there are explicit causal lexical markers, and other sort of expressions that have a discourse meaning,

| Corpus | Version | Causal | Non-Causal | Total |
|---|---|---|---|---|
| Training | non bootstrapping | 7,606 | 79,290 | 86,896 |
| | bootstrapping | 12,534 | 88,210 | 100,744 |
| Development | | 181 | 307 | 488 |
| Test | | 315 | 296 | 611 |

Table 3: Number of instances in the AltLex corpus

which are called AltLex. The relations signalled by an AltLex expression are implicit relations, in which the annotator did not find an appropriate connective to insert between the events, because the meaning of the relation is entailed by other expressions, namely AltLex. Furthermore, some causal lexical markers can be perceived as variations of the basic ones, but those variations are not usually in the list of discourse connectives. Thus, the authors of the AltLex corpus decided to develop a method to identify a larger amount of AltLex expressions with causal meaning. The corpus construction leveraged Simple Wikipedia, by aligning sentences from Wikipedia that consist of unknown lexical causal markers with sentences from Simple Wikipedia that contain corresponding known lexical causal markers. Once a first set of causal and non-causal sentences were identified, a *bootsrapping* method was applied to enhance the corpus. We call "non bootstrapping" to the first version of the corpus and "bootstrapping" to the second one. The corpus size is shown in Table 3. For further details see (Hidey and McKeown, 2016).

The class distribution of the lexical markers should be studied, because one of the features of the corpus is the annotation of the lexical markers susceptible of expressing causation. Table 4 shows the class distribution of the lexical markers, as well as the the number of unarguable ones and the number of lexical markers with mostly a different meaning in the training and the test set. According to the Table 4, there are few ambiguous connectives, 121 in the "non bootstrapping" corpus and 147 in the "bootstrapping" version. However, there is an important difference between the two versions of

|  | Bootstrapping | |
| --- | --- | --- |
|  | No | Yes |
| Total | 8214 | 8854 |
| Unambiguous in `Causal` class | 922 | 1034 |
| Unambiguous in `Non Causal` class | 7171 | 7673 |
| `Causal` in train, `Non Causal` in test | 0 | 27 |
| `Non Causal` in train, `Causal` in test | 0 | 8 |

Table 4: Distribution of the lexical markers

the corpus: there are no differences between the training and the test set in the "non bootstrapping" version, but there are in the "bootstrapping" one. This fact means that the "bootstrapping" version of the corpus has some instances that do not follow the class distribution of the training set, so they represent a higher difficulty for the classification system. Section 4.3 exposes the good performance of the proposed system in some of those instances.

### 3.3 Disambiguation of the Causal Meaning

Kruengkrai et al. (2017) describe a Multi-column Convolutional Neural Network (CNN) that uses background knowledge for the identification of the causal meaning of an input sentence in Japanese. Convolutional networks excel at pattern learning in input data, however causation does not have a particular syntactic structure as it was mentioned. Moreover, CNN requires the definition of the kernel size, which means that we should know beforehand the length of the context that projects the causal meaning of the sentence. Nevertheless, the list of expressions that may represent a causal meaning is not limited and some sentences implicitly express causality. On the other hand, Long short-term memory (LSTM) recurrent neural networks (RNN) (Hochreiter and Schmidhuber, 1997) can encode the sequential information of the input text. Melamud et al. (2016) also showed that using an LSTM results in a more precise context encoding and substantially improve performance in several tasks upon the common average-of-words-embeddings representation. Therefore, we propose the use of LSTM RNN instead of a CNN, because we argue that LSTM has a higher capacity of encoding meaning for the task of causality classification.

We propose a neural network architecture with two inputs, which is mainly based on the encoding of the two inputs with an LSTM and the use of several dense layers with a *tanh* activation function.

Following the assumption that some sort of relation should exist between the two events of a causal relation, the first evaluated model consists of two

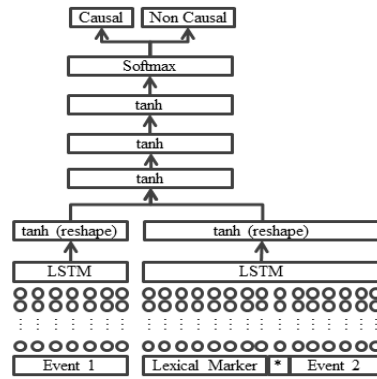

Figure 1: The network architecture

connected LSTMs, or in other words, the LSTM network of the second event of the causal relation is initialized with the last state of the first LSTM. We call to this architecture "Stated_Pair_LSTM". We also evaluated the same architecture but without the connection between the two LSTMs. We call to this second architecture "Pair_LSTM" and it is depicted in Figure 1.

According to Figure 1, the first input of the system is the first event of the relation, and the second input is the concatenation of the lexical marker and the second event. The text processing starts with the *tokenization* of the two inputs, and the representation of them as a matrix of word embedding vectors. The set of pre-trained embeddings used was Glove (Pennington et al., 2014), specifically the 300 dimension reference vectors of the 840B cased tokens set.

The lengths of the first ($n$) and the second ($m$) input are not the same, so three zero-padding strategies were evaluated to measure which of them is the most convenient to encode the causal meaning. The maximum, the mean and mode of the length of the two inputs were calculated:

$$\forall x \in \mathbb{R}^{n \times d} \text{ and } \forall y \in \mathbb{R}^{m \times d}$$
$$pad(x) = \{max(x), mean(x), mode(x)\} \quad (1)$$

where $x$ and $y$ are the first and second input and $d$ is the length of the word embedding vectors.

Subsequently, each of the two outputs of the encoding layer are vectorized to a vector of length 100 by a dense layer with a *tanh* activation function. The context of the causal relation is represented by the concatenation of the two vectors.

$$\forall W \in \mathbb{R}^{100 \times nd} \text{ and } \forall b \in \mathbb{R}^{100}$$
$$vec(x) : \mathbb{R}^{n \times d} \to \mathbb{R}^{nd}$$
$$tanh(W \cdot x + b) \quad (2)$$
$$context : (vec(x), vec(y))$$

The output of the Equation 2 is processed by three dense layers activated by a *tanh* function. The last layer is composed of the *softmax* operation.

The performance of two learning optimizers was evaluated, specifically Adadelta (Zeiler, 2012) and Adam (Kingma and Ba, 2015). In order to avoid the overfitting of the classification process, different values for dropout ($[0.5, 0.75]$) and regularization were used ($\left[8 \cdot 10^{-3}, 8 \cdot 10^{-6}\right]$).

## 4 Experiments and Results

As far as we know, the AltLex corpus has been only used in the work in which it was presented (Hidey and McKeown, 2016), therefore we consider their method as the state of the art on that corpus. Moreover, as it has been mentioned several times, the task of causality classification lacks of representative or large enough corpus that covers a wide range of causal expressions, as well as without the restriction of being composed of specific causal constructions. Therefore, the methods were only evaluated with the AltLex corpus. We have used the two versions of the corpus ("non bootstrapping" and "bootstrapping") in our experiments. The size of each of them is shown in Table 3.

### 4.1 Baselines

We consider two baselines in order to compare our proposal. The first one (B1) is founded on the use of the most common class of each causal connective according to its class in the training data. For example, if the lexical marker *since* has mostly a causal meaning, B1 always classifies the relation signalled by *since* as `Causal`. A similar approach was also used in (Hidey and McKeown, 2016).

The second baseline (B2) is the system of Hidey and McKeown (2016), which is based on SVM with a large set of features generated from the combination of features from the original parallel corpus and some lexical resources (WordNet, Verb-Net and PropBank). The fact of relying on lexical resources restricts the recall of the system to the linguistic coverage of the lexical resources. In contrast, we propose a neural network architecture fed only by a set of word embedding vectors, which has a higher ability of generalization as it is shown later in the paper.

### 4.2 Results

The results reached by our system are shown in Table 5.[1] The Precision, Recall and F1 values were used to measure the performance of the system in the `Causal` Class, and the Accuracy to measure the overall performance of the system.

The performance of B1 defines a hard baseline for the two versions of the corpus. On the other hand, the good performance of B1 might indicate that the training corpus is composed of few unambiguous causal connectives. Concerning the results reached by the proposed systems, all the configurations in Table 5 outperform the baseline B1, which means that the system learns beyond the class distribution of the lexical markers in the training data.

Those configurations that use Adadelta as optimizer outperform the system B2 in the "non bootstrapping" version of the corpus. The behaviour of B2 shows a big difference between Precision and Recall, indicating that the system has many false positives, i.e. a large number of sentences without a causal meaning. This is not a desirable behaviour. On the other hand, our system has a high Precision with lower Recall, indicating it mainly classifies correctly sentences with unambiguous connectives. Figure 2 shows that the most frequent connectives in the `Causal` class do not have any instance in the `Non Causal` class, which supports our last assertion. The best configuration ("Pair_LSTM_ Max_Adadelta") uses the maximum operation for the zero-padding strategy and Adadelta as learning optimizer. Our proposal in this scenario improves the state of the art by 2.13% according to F1 in the Causal class (C) and 7.72% according to Accuracy.

We observe a similar trend in the performance of the system with the "bootstrapping" version of the corpus, i.e. the system architecture "Pair_LSTM" tends to reach better results than the architecture "Stated_LSTM". This behaviour means that our assumption about the relation between the meanings of the two input events does not hold, so it is better to encode each argument independently and then to measure the relation between the arguments by using dense layers. Concerning the optimizers, the results show a similar behaviour in both architectures. The Adadelta optimizer promotes the Recall and penalizes Precision, whereas the system reaches a higher value of Precision and a smaller difference between Precision and Recall

---

[1] For the sake of brevity, those systems that performed worse than B1 and B2 are not in Table 5.

| Corpus | Method | Precision C. | Recall C. | F1 C. | Accuracy |
|---|---|---|---|---|---|
| Non bootstrapping | B1 | 68.92% | 54.92% | 61.13% | 63.99% |
| | B2 | 70.28% | 77.60% | 73.76% | 71.86% |
| | Stated_Pair_LSTM_Mean_Adadelta | 90.04% | 60.31% | 72.24% | 76.10% |
| | Stated_Pair_LSTM_Mode_Adadelta | 89.23% | 63.17% | 73.97% | 77.08% |
| | **Pair_LSTM_Max_Adadelta** | 88.46% | 65.71% | **75.40%** | **77.90%** |
| | Pair_LSTM_Mean_Adadelta | 89.33% | 63.80% | 74.44% | 77.41% |
| Bootstrapping | B1 | 74.38% | 86.66% | 80.05% | 77.74% |
| | B2 | 77.29% | 84.85% | 80.90% | 79.58% |
| | Stated_Pair_LSTM_Max_Adadelta | 78.69% | 84.44% | 81.47% | 80.19% |
| | Stated_Pair_LSTM_Mean_Adam | 80.00% | 82.53% | 81.25% | 80.36% |
| | Stated_Pair_LSTM_Mode_Adam | 80.24% | 82.53% | 81.37% | 80.52% |
| | Pair_LSTM_Max_Adadelta | 78.07% | 84.76% | 81.12% | 79.86% |
| | Pair_LSTM_Mean_Adadelta | 78.48% | 85.71% | 81.94% | 80.52% |
| | **Pair_LSTM_Mean_Adam** | **80.30%** | 82.85% | 81.56% | **80.68%** |
| | Pair_LSTM_Mode_Adam | 77.24% | 87.30% | 81.96% | 80.19% |

Table 5: Results of the baselines (B1 and B2) and the configurations evaluated

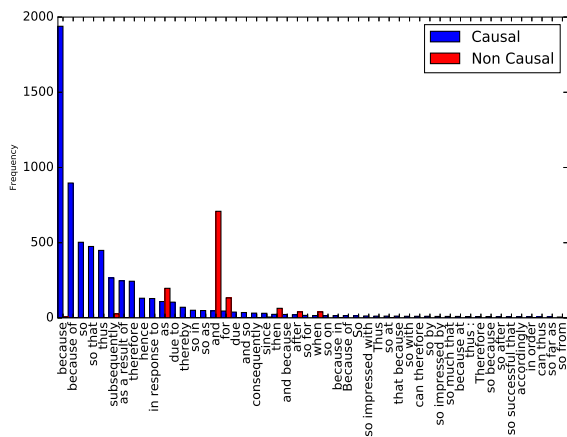

Figure 2: Distribution of the most frequent connectives in the class Cause

when Adam optimizer is used, which is a more desirable behaviour. A high value of Precision and a high value of Recall mean that there is a good balance between the accuracy in the disambiguation of the causal meaning of the relation and the coverage of different ways of expressing causality. To conclude, the system that reached the highest performance is the one that used the mean strategy for zero-padding and the optimizer Adam.

Most of the configurations of the proposed neural network outperform the baseline B2. If we look at the values of Precision, our proposal improves the Precision value of B2 by 3.89% . However, B2 is better 2.41% according to the Recall value. The good Recall value of B2 system entails a drop in the performance according to the Precision. This behaviour is also developed by the neural network configurations that use maximum strategy for zero-padding and Adadelta as optimizer, but in those cases the Recall and the Pre-

cision are greater. As previously mentioned, this means that the systems have a large number of false positive instances, so they might not be correctly learning the characteristics of the Causal class. In contrast, "Pair_LSTM_Mean_Adam" reached the best Precision in the Causal class and the less difference between Precision and Recall, thus it also reached a better balance between the accuracy in the disambiguation of the causal relation and the coverage of different ways of expressing causality.

| Corpus | Version | Causal | Non Causal | Total |
|---|---|---|---|---|
| Training | Non bootstrapping | 7,606 | 7,929 | 15,534 |
| | Bootstrapping | 12,534 | 8,821 | 21,354 |

Table 6: Number of relations in the reduced version of the AltLex Corpus

The class distribution of the AltLex corpus is not balanced (see Table 3), so we reduced the number of instances of the Non Causal by a factor of ten with the aim of evaluating our system with a more balanced dataset. The new corpus size is in Table 6. We assessed the performance of the baseline B1 and the system "Pair_LSTM_Mean_Adam" and the results are in Table 7. Regarding the performance of B1 in the "non bootstrapping version", the Precision is lower and the Recall is higher than in the previous experiments, which means that the number of false positives is large, so it is not learning the characteristics of the class Causal. As in the previous set of experiments, B1 promotes Recall and penalizes Precision in the "bootstrapping" version of the corpus, which is not a desirable behaviour. In contrast, our network architecture followed the same trend in both set of experiments and with the both version of the corpus. "Pair_LSTM_Mean_Adam"

| Corpus | Method | Precision C. | Recall C. | F1 C. | Accuracy |
|---|---|---|---|---|---|
| Non bootstrapping | B1 | 63.70% | 84.12% | 72.50% | 67.10% |
| | **Pair_LSTM_Mean_Adam** | **76.38%** | 69,84% | **72,96%** | **73.32%** |
| Bootstrapping | B1 | 67.34% | 94.28% | 78.57% | 73.48% |
| | **Pair_LSTM_Mean_Adam** | **72.65%** | 88.57% | **79.82%** | **76.92%** |

Table 7: Results of B1 and Pair_LSTM_Mean_Adam with the reduced version of the AltLex corpus

| Sentence | Training | Test | B2 | Our proposal |
|---|---|---|---|---|
| The United States decided to *break* off economic relations with Cuba (which means that they would stop buying things from them). | Causal | Non Causal | Causal | Non Causal |
| Although Roosevelt had promised *to keep* the United States out of the war, he nevertheless took concrete steps to prepare for war. | Causal | Non Causal | Causal | Non Causal |
| Greatly alarmed and with Hitler *making* further demands on the Free City of Danzig, Britain and France guaranteed their support for Polish independence; when Italy conquered Albania in April 1939, the same guarantee was extended to Romania and Greece. | Causal | Non Causal | Causal | Non Causal |
| One of these fragments gives rise to fibrils of amyloid beta, *which then* form clumps that deposit outside neurons in dense formations known as senile plaques. | Ambiguous | Causal | Non Causal | Causal |

Table 8: Some correctly classified examples by our best configuration and misclassified by B2

significantly improves the performance of B1 according to the McNemar's test with a P-value of 0.005 and 0.05 respectively.

### 4.3 Analysis

As mentioned in the previous sections, there are some linguistic constructions that can represent a causal meaning in some contexts, but not in other and vice-versa. A better balance between Precision and Recall in the Causal class means that the system learns the causal meaning better. Roughly speaking, the quality of the causal disambiguation is better, also for those lexical markers that are mostly Causal in the training data. We present some correctly classified examples by our best configuration and misclassified by B2 in Table 8. It shows the class of the instance in the test corpus and the value of the most frequent class in the training corpus. The behaviour of the verb *break* should be stressed since it is considered as a causative verb. However, there are some other uses of *break* that do not have a causal meaning (see Table 8).

The behaviour of the proposed system with the connective *which then* is also remarkable. That connective is totally ambiguous, because it only has one instance labelled as Causal and other as Non Causal in the training data. We have to take into account that we work with word embeddings, so if the individual words *which* and *then* are over-represented in one class, that fact can determine the behaviour of the classification system.

The individual term *which* does not appear in any instance of the training corpus, while the word *then* is mostly in sentences without a causal meaning. Despite the difficulty of the expression *which then*, we can see in Table 8 that our proposed system correctly disambiguate its causal meaning, while the system of Hidey and McKeown (2016) does not. Therefore, the results reached and those examples allow us to confirm our claim that the encoding of the context is required for the disambiguation of the causal meaning as we have shown in this paper.

## 5 Conclusions and Future Work

We defined the task of causation classification as a task composed of another two subtasks: causal meaning classification and causal argument classification. The paper was focused on the first subtask, and we claim that the encoding of the two events of the relation is required for a suitable disambiguation of causality. We proposed an encoding system based on a neural network with two inputs, one for the first event and the other for the lexical marker and the second event. Our proposed system outperforms the state-of-the-art. We also showed the success of the system in some non-causative sentences but with commonly causative verbs (see Table 8).

One of the problems of the task is the lack of resources, so, as future work, we plan the creation of a new corpus for the two subtasks of causality classification, namely causality disambiguation and causality argument classification.

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
