# Peer review of "Neural Disambiguation of Causal Lexical Markers based on Context"

_ACL 2017 — decision unknown_

[Official Review · Reviewer 1 · rating 2 · confidence 4]
soundness 5 · originality 5 · clarity 4 · impact 3 · substance 4 · appropriateness 5 · meaningful comparison 3 · presentation format Poster

This paper develops an LSTM-based model for classifying connective uses for
whether they indicate that a causal relation was intended. The guiding idea is
that the expression of causal relations is extremely diverse and thus not
amenable to syntactic treatment, and that the more abstract representations
delivered by neural models are therefore more suitable as the basis for making
these decisions.

The experiments are on the AltLex corpus developed by Hidley and McKeown. The
results offer modest but consistent support for the general idea, and they
provide some initial insights into how best to translate this idea into a
model. The paper distribution includes the TensorFlow-based models used for the
experiments.

Some critical comments and questions:

* The introduction is unusual in that it is more like a literature review than
a full overview of what the paper contains. This leads to some redundancy with
the related work section that follows it. I guess I am open to a non-standard
sort of intro, but this one really doesn't work: despite reviewing a lot of
ideas, it doesn't take a stand on what causation is or how it is expressed, but
rather only makes a negative point (it's not reducible to syntax). We aren't
really told what the positive contribution will be except for the very general
final paragraph of the section.

* Extending the above, I found it disappointing that the paper isn't really
clear about the theory of causation being assumed. The authors seem to default
to a counterfactual view that is broadly like that of David Lewis, where
causation is a modal sufficiency claim with some other counterfactual
conditions added to it. See line 238 and following; that arrow needs to be a
very special kind of implication for this to work at all, and there are
well-known problems with Lewis's theory (see
http://bcopley.com/wp-content/uploads/CopleyWolff2014.pdf). There are comments
elsewhere in the paper that the authors don't endorse the counterfactual view,
but then what is the theory being assumed? It can't just be the temporal
constraint mentioned on page 3!

* I don't understand the comments regarding the example on line 256. The
authors seem to be saying that they regard the sentence as false. If it's true,
then there should be some causal link between the argument and the breakage.
There are remaining issues about how to divide events into sub-events, and
these impact causal theories, but those are not being discussed here, leaving
me confused.

* The caption for Figure 1 is misleading, since the diagram is supposed to
depict only the "Pair_LSTM" variant of the model. My bigger complaint is that
this diagram is needlessly imprecise. I suppose it's okay to leave parts of the
standard model definition out of the prose, but then these diagrams should have
a clear and consistent semantics. What are all the empty circles between input
and the "LSTM" boxes? The prose seems to say that the model has a look-up
layer, a Glove layer, and then ... what? How many layers of representation are
there? The diagram is precise about the pooling tanh layers pre-softmax, but
not about this. I'm also not clear on what the "LSTM" boxes represent. It seems
like it's just the leftmost/final representation that is directly connected to
the layers above. I suggest depicting that connection clearly.

* I don't understand the sentence beginning on line 480. The models under
discussion do not intrinsically require any padding. I'm guessing this is a
requirement of TensorFlow and/or efficient training. That's fine. If that's
correct, please say that. I don't understand the final clause, though. How is
this issue even related to the question of what is "the most convenient way to
encode the causal meaning"? I don't see how convenience is an issue or how this
relates directly to causal meaning.

* The authors find that having two independent LSTMs ("Stated_LSTM") is
somewhat better than one where the first feeds into the second. This issue is
reminiscent of discussions in the literature on natural language entailment,
where the question is whether to represent premise and hypothesis independently
or have the first feed into the second. I regard this as an open question for
entailment, and I bet it needs further investigation for causal relations too.
So I can't really endorse the sentence beginning on line 587: "This behaviour
means that our assumption about the relation between the meanings of the two
input events does not hold, so it is better to encode each argument
independently and then to measure the relation between the arguments by using
dense layers." This is very surprising since we are talking about subparts of a
sentence that might share a lot of information.

* It's hard to make sense of the hyperparameters that led to the best
performance across tasks. Compare line 578 with line 636, for example. Should
we interpret this or just attribute it to the unpredictability of how these
models interact with data?

* Section 4.3 concludes by saying, of the connective 'which then', that the
system can "correctly disambiguate its causal meaning", whereas that of Hidey
and McKeown does not. That might be correct, but one example doesn't suffice to
show it. To substantiate this point, I suggest making up a wide range of
examples that manifest the ambiguity and seeing how often the system delivers
the right verdict. This will help address the question of whether it got lucky
with the example from table 8.

[Official Review · Reviewer 2 · rating 3 · confidence 4]
soundness 5 · originality 5 · clarity 5 · impact 3 · substance 4 · appropriateness 5 · meaningful comparison 3 · presentation format Oral Presentation

This paper proposes a method for detecting causal relations between clauses,
using neural networks ("deep learning", although, as in many studies, the
networks are not particularly deep).  Indeed, while certain discourse
connectives are unambiguous regarding the relation they signal (e.g. 'because'
is causal) the paper takes advantage of a recent dataset (called AltLex, by
Hidey and McKeown, 2016) to solve the task of identifying causal vs. non-causal
relations when the relation is not explicitly marked.  Arguing that
convolutional networks are not as adept as representing the relevant features
of clauses as LSTMs, the authors propose a classification architecture which
uses a Glove-based representation of clauses, input in an LSTM layer, followed
by three densely connected layers (tanh) and a final decision layer with a
softmax.

The best configuration of the system improves by 0.5-1.5% F1 over Hidey and
MCkeown's 2016 one (SVM classifier).  Several examples of generalizations where
the system performs well are shown (indicator words that are always causal in
the training data, but are found correctly to be non causal in the test data).
Therefore, I appreciate that the system is analyzed qualitatively and 
quantitatively.

The paper is well written, and the description of the problem is particularly
clear. However a clarification of the differences between this task and the 
task of implicit connective recognition would be welcome.  This could possibly 
include a discussion of why previous methods for implicit connective 
recognition cannot be used in this case.

It is very appreciable that the authors uploaded their code to the submission
site (I inspected it briefly but did not execute it).  Uploading the (older)
data (with the code) is also useful as it provides many examples.  It was not
clear to me what is the meaning of the 0-1-2 coding in the TSV files, given
that the paper mentions binary classification. I wonder also, given that this
is the data from Hidey and McKeown, if the authors have the right to repost it
as they do.  -- One point to clarify in the paper would be the meaning of
"bootstrapping", which apparently extends the corpus by about 15%: while the
construction of the corpus is briefly but clearly explained in the paper, the
additional bootstrapping is not. 

While it is certainly interesting to experiment with neural networks on this
task, the merits of the proposed system are not entirely convincing.  It seems
indeed that the best configuration (among 4-7 options) is found on the test
data, and it is this best configuration that is announced as improving over
Hidey by "2.13% F1".  However, a fair comparison would involve selecting the
best configuration on the devset.

Moreover, it is not entirely clear how significant the improvement is. On the
one hand, it should be possible, given the size of the dataset, to compute some
statistical significance indicators.  On the other hand, one should consider
also the reliability of the gold-standard annotation itself (possibly from the
creators of the dataset).  Upon inspection, the annotation obtained from the
English/SimpleEnglish Wikipedia is not perfect, and therefore the scores might
need to be considered with a grain of salt.

Finally, neural methods have been previously shown to outperform human
engineered features for binary classification tasks, so in a sense the results 
are rather a confirmation of a known property. It would be interesting to see
experiments with simpler networks used as baselines, e.g. a 1-layer LSTM.  The
analysis of results could try to explain why the neural method seems to favor 
precision over recall.